# New models for donor-recipient matching in lung transplantations

J. M. Dueñas-Jurado[1]ᵒ, P. A. Gutiérrez[2], A. Casado-Adam[3]*, F. Santos-Luna[4]ᵒ, A. Salvatierra-Velázquez[5]¤, S. Cárcel[1,6]‡, C. J. C. Robles-Arista[1]‡, C. Hervás-Martínez[2]‡

**1** Intensive Care Unit, Reina Sofia University Hospital, Cordoba, Spain, **2** Department of Computer Science and Numerical Analysis, Universidad de Córdoba, Córdoba, Spain, **3** General and Digestive Surgery Unit, Reina Sofia University Hospital, Cordoba, Spain, **4** Pneumology and Lung Transplant Service, Reina Sofia University Hospital, Cordoba, Spain, **5** Thoracic Surgery and Lung Transplantation Service, Reina Sofia University Hospital, Cordoba, Spain, **6** Maimonides Institute for Research in Biomedicine of Cordoba (IMIBIC), Cordoba, Spain

ᵒ These authors contributed equally to this work.
¤ Current address: Department of Surgery, Universidad de Córdoba, Córdoba, Spain
‡ SC, CJCRA and CHM also contributed equally to this work.
* angeti@hotmail.com

**Data Availability Statement:** All relevant data are within the paper and its Supporting Information files.

## Abstract

### Objective

One of the main problems of lung transplantation is the shortage of organs as well as reduced survival rates. In the absence of an international standardized model for lung donor-recipient allocation, we set out to develop such a model based on the characteristics of past experiences with lung donors and recipients with the aim of improving the outcomes of the entire transplantation process.

### Methods

This was a retrospective analysis of 404 lung transplants carried out at the Reina Sofía University Hospital (Córdoba, Spain) over 23 years. We analyzed various clinical variables obtained via our experience of clinical practice in the donation and transplantation process. These were used to create various classification models, including classical statistical methods and also incorporating newer machine-learning approaches.

### Results

The proposed model represents a powerful tool for donor-recipient matching, which in this current work, exceeded the capacity of classical statistical methods. The variables that predicted an increase in the probability of survival were: higher pre-transplant and post-transplant functional vital capacity (FVC), lower pre-transplant carbon dioxide (PCO2) pressure, lower donor mechanical ventilation, and shorter ischemia time. The variables that negatively influenced transplant survival were low forced expiratory volume in the first second (FEV1) pre-transplant, lower arterial oxygen pressure (PaO2)/fraction of inspired oxygen (FiO2) ratio, bilobar transplant, elderly recipient and donor, donor-recipient graft disproportion requiring a surgical reduction (Tailor), type of combined transplant, need for

**Funding:** This study was funded by the Spanish Ministry of Economy and Competitiveness (MINECO), and FEDER funds (EU) to PAG and CHM (TIN2017-85887-C2-1-P and TIN2017-90567-REDT).

**Competing interests:** The authors have declared that no competing interests exist.

cardiopulmonary bypass during the surgery, death of the donor due to head trauma, hospitalization status before surgery, and female and male recipient donor sex.

## Conclusions

These results show the difficulty of the problem which required the introduction of other variables into the analysis. The combination of classical statistical methods and machine learning can support decision-making about the compatibility between donors and recipients. This helps to facilitate reliable prediction and to optimize the grafts for transplantation, thereby improving the transplanted patient survival rate.

## Introduction

In the last 30 years, lung transplantation has become the only therapeutic option for patients in respiratory failure who have exhausted other medical and surgical treatments. Thus, last year 2019, 419 lung transplants were performed in Spain, accounting for a total of 5,239 lung transplants performed since 1990 in Spain [1]. Despite the accumulated experience, immunological advances, surgical techniques, and management in the immediate postoperative period, the mortality and morbidity associated with this type of transplant surgery is higher than that of other organs, especially in the first few months.

Several factors limit lung transplant survival. Among the most prominent in the long term are recipient age and the presence of respiratory diseases [2]. Therefore, it is important to unify criteria for selecting donors and candidates for lung transplantation in order to advance our understanding of the evolution of lung transplants. This could include the use of predictive methods [3] or the early identification of comorbidities in patients—a factor that influences survival both before and after transplantation. Many variables must be taken into account when performing a lung transplant and can influence the surgical outcome, such as age, anatomical and immunological compatibility between donor and recipient, transplant organ conservation and status, surgery type, and transplantation type. As a consequence, multidisciplinary teams are responsible for managing and evaluating the transplantation process [4].

Two different problems are classically associated with lung transplants, namely organ shortages and donor-recipient allocation. In terms of the former, of the 2,301 multiorgan donors that were offered in Spain in 2019, only 690 were offered as lung donors and only 392 were transplanted [1]. Compared to other organs such as the kidney or liver, the number of lungs suitable for transplantation is limited by certain characteristics. This organ must comply with a series of variables such as, for example, a donor oxygenation index—known as the arterial oxygen pressure (PaO2) to inspired oxygen fraction (FiO2) ratio—exceeding 300, age under 55 years, radiologically normal organ images, etc.

One of the main challenges for professionals specializing in the management of organ donors is the maintenance of the lung donor so as to optimize and validate the donated lungs, which requires the application of specific protocols in multi-organ donors for lung donation. The following metrics are associated with lung validity for transplantation, without affecting survival or primary graft failure: (1) an apnea test with positive pressure;(2) mechanical ventilation with 6–8 ml/kg tidal volume and a positive end-expiration pressure of 8–10 cm of water; (3) recruitment maneuvers; (4) bilateral bronchoscopic lavage procedures; (5) alveolar recruitment maneuvers;(6) monitoring and management of hemodynamic parameters with a central

venous pressure (PVC) <8 cm of water; and whenever possible, (7) an extravascular lung water index (ELWI) <10 ml/kg [5]. The application of these protocols does not affect the maintenance, number of valid organs, or transplantation or functioning of other extrapulmonary organs compared to other intensive treatments [6].

The imbalance between lung supply and donor recipient demand has remained relatively stable in recent years. This is because, although the number of lung donors has grown in recent years, indications for transplantation have also increased, making the waiting list for transplants difficult to reduce. Thus, in Spain in 2019, although 419 transplants were performed, 473 new patients were put on the waiting list, meaning that 258 recipients remained on the lung transplant waiting list at the end of 2019, in the context of 2% risk of mortality for the patients on this list [1].

Donor-recipient lung allocation is either based on the experience of the resident pulmonologists or thoracic surgeons or, especially in North America, transplant teams follow the lung allocation score (LAS) method for donor-recipient allocation. The LAS was created in May 2005 and is used as a predictive model of morbidity and mortality in lung transplantation. However, contrary to initial assumptions, it has not been internationally accepted because it has several limitations, such as not considering important donor and recipient parameters [7, 8]. Therefore, here we have reviewed the allocation of organs to try to identify emerging areas of knowledge that can help improve and optimize lung grafts, reduce hospital morbidity and mortality, and enhance the general results.

## Materials and methods

When trying to optimize lung transplantation donor-recipient allocation, the following principles must be considered (1) Justice: equal access and fair allocation of the organs obtained for transplantation must be guaranteed; this can be influenced by several factors, including donor and recipient age, clinical urgency for the transplant, time on the waiting list, immunological characteristics, among others. (2) Efficiency: which can be conceptualized as taking full advantage of limited resources and avoiding their misuse [9]; for example, most allocation models do not prioritize patients who will consume fewer resources, have a shorter hospital stay, longer survival time, etc. (3) Utility: understood as maximization of the general desired good, based on survival of the recipients and transplanted lungs, recipient quality of life, and availability of alternative treatments [10].

In areas with low donation rates, the volume of lung transplants may decrease while the waiting list times increase. This means that maximization of the general use of available organs, while still maintaining a fair and equitable allocation system is an even higher priority, perhaps requiring revision of the existing allocation criteria. Thus, we searched for potential new assignment models based on the classical statistical methodology associated with artificial intelligence known as 'neural networks'. In this work we aimed to (a) study the predictability of survival at six months using a homogeneous data set where all the transplants had been performed in the same hospital; (b) see if the use of machine learning could outperform traditional statistical approaches to predict mortality at six months and thus, create a powerful statistical model with advantages over the current organ allocation systems.

## Statistical methods

We combined logistic regression (LR) with a special type of neural network in order to obtain the advantages of both. LR is a widely used and accepted method of analyzing binary or multiclass outcome variables, which has the flexibility for use to predict the probability state of a dichotomous variable based on predictive variables. In turn, machine learning methods

comprise different supervised training algorithms and their implementation has led to significant improvements over classical statistical methods in many fields of application. Both machine learning and LR methods were used for different purposes to analyze the lung transplants in this current work [11–13]. First we carried out binary comparisons (survival/non-survival) and then we extended the same methodology to multiclass problems [14].

We applied evolutionary algorithms (EAs) and product unit neural networks (PUNNs) to increase the precision of our vector architecture design. This resulted in an evolutionary PUNN (EPUNN) which optimized the result, taking it from a local search to more globalized optimization of the results [15]. Finally, by combining LR with EPUNN, we obtained better quality binary classifiers [16]. This hybrid perspective first determines an optimized EPUNN model by performing a global search, both in terms of architecture and weights [17]. Using LR initial covariates and product units (LRIPU), our model achieves stable and competitive precision with a relatively simple structure, thereby facilitating its application and interpretation of the results.

## Data collection

The database we used contained data from a total of 404 lung transplant patients from the time the Lung Transplant Program was created at the Reina Sofía University Hospital (Córdoba, Spain) in October 1993, up until January 2016. The lung donors had come from any of the 185 hospitals accredited for donation in Spain. Pediatric recipients (<15 years) and combined or multi-organ transplants (of more than two vital organs) were excluded from the study. Follow-up was carried out during the first post-transplantation year or until the time of death or graft loss. Different recipient and donor characteristics and intraoperative and postoperative factors had been included for each of these transplants, as summarized in Table 1; categorical variables were transformed into binary variables for processing with the different models.

## Results

This work allowed us to obtain the optimal LRIPU model, as shown in Table 2. The model has 48 weights and a hybrid structure: the non-linear part includes a product unit base function with 11 coefficients (obtained by the EPUNN method), while the linear one was based on 37 coefficients associated with the 36 donor-recipient pairing characteristic input variables (Table 1) plus the bias term. The reasonable size of the LRIPU model allows the coefficients of the model to be studied, which would be impossible for other next-generation non-linear methods. The dependent variable was graft survival, when considering the end-point at six months.

To analyze the relative importance of each covariate over the target variable, we focused on the probabilities associated with the covariate when the rest equaled zero. Thus, any variables with high positive values will have odds lower than 1, thereby increasing the probability of survival at six months when the value of said variable increases. This was the case for variables x1, x3, x5, x6, x8, x15, x16, x17, x18, x21, x22, x23, x25, x26, x28, x31, x32, x33, and x34, with x33 (fvc_pre) showing the strongest influence and predicting the highest coefficient (23.50).The percentage of FVC in patients with various respiratory diseases referred for lung transplantation evaluation was an independent predictor of mortality at one year or for the need for transplantation [18]; x34 (fvc_prep, 3.03) FVC was related to transplant survival, with a decrease in survival after surgery shown in 42% of patients who do not normalize these parameters after transplantation [19], which was followed by x32 (pco2_pre, 3.12).

Variations in $CO_2$, even in patients who had already received a lung transplant, subtly modify breathing control so that in COPD patients with elevated $CO_2$ levels, these levels did not

**Table 1. List of the variables used in the study along with a description.**

| Variable | Description |
|---|---|
| $x_1$ (sex_rec) {1,0} | Recipient sex: male (1), female (0) |
| $x_2$ (age_rec) | Recipient age at the time of transplant. |
| $x_3$ (desease2 = COPD) | Recipient pathology: chronic obstructive pulmonary disease (COPD), cystic fibrosis (CF), pulmonary fibrosis (PF), bronchiectasis, or others. |
| $x_4$ (desease2 = PF) | |
| $x_5$ (desease2 = bronchiectasis) | |
| $x_6$ (desease2 = CF) | |
| $x_7$ (desease2 = others) | |
| $x_8$ (pre_tx = ambulatory) | Pre-transplant recipient status: ambulatory, hospitalized, or in an intensive care unit (ICU) |
| $x_9$ (pre_tx = hospitalized) | |
| $x_{10}$ (pre_tx = ICU) | |
| $x_{11}$ (sex_donor) {1,0} | Donor sex: male (1), female (0) |
| $x_{12}$ (age_donor) | Donor age |
| $x_{13}$ (death_d = HS) | Cause of death: hemorrhagic stroke (HS), traumatic brain injury (TBI), anoxia, ischemic stroke (IS), or other. |
| $x_{14}$ (death_d = TBI) | |
| $x_{15}$ (death_d = other) | |
| $x_{16}$ (death_d = ANOXIA) | |
| $x_{17}$ (death_d = IS) | |
| $x_{18}$ (ti_in_do) | Time with endotracheal intubation (days). |
| $x_{19}$ (io2_donor) | Index of donor oxygenation: arterial oxygen blood pressure (PaO2)/inspired fraction of oxygen (FiO2). |
| $x_{20}$ (type_tx = singleft) | Type of transplant: single left or right lung transplant, bilateral lung transplant, bilobular cadaver. |
| $x_{21}$ (type _tx = singright) | |
| $x_{22}$ (type _tx = bilateral) | |
| $x_{23}$ (type _tx = bilobular cadaver) | |
| $x_{24}$ (type _tx = Ho+Bipulm) | |
| $x_{25}$ (Cold ischemia time = Long) | Cold ischemia time: short (0 to 2 hours), medium (2 to 4 hours), long (4 to 6 hours), and very long (>6 hours). |
| $x_{26}$ (Cold ischemia time = Medium) | |
| $x_{27}$ (Cold ischemia time = Very long) | |
| $x_{28}$ (Cold ischemia time = Short) | |
| $x_{29}$ (bypass) {1, 0} | Application of a cardiopulmonary bypass: 1 (YES), 0 (NO). |
| $x_{30}$ (Tailor) {1, 0} | Pulmonary tailoring: 1 (YES), 0 (NO). |
| $x_{31}$ (po2_pre) | Oxygen pressure (PO2) pre-transplant. |
| $x_{32}$ (pco2_pre) | Carbon dioxide pressure (PCO2) pre-transplant. |
| $x_{33}$ (fvc_pre) | Functional vital capacity (FVC) pre-transplant. |
| $x_{34}$ (fvc_prp) | Functional vital capacity (FVC) post-transplant. |
| $x_{35}$ (fev1_pre) | Forced expiratory volume in first second (FEV1) pre-transplant. |
| $x_{36}$ (fev1_prp) | Forced expiratory volume in first second (FEV1) post-transplant. |
| $y$ (Survival) {1, 0} | Survival of the graft after six months: 1 (YES), 0 (NO). |

normalize until the third week after surgery [20]. It was already known that hospitalization for hypercapnia greater than 50 mmHg is a poor prognostic criteria in patients with chronic obstructive pulmonary disease, resulting in decreased survival rates [21]. We found that x18 (ti_in_do) had a minor but significant influence on increased mechanical ventilation times in donors which, in turn, favors the appearance of primary lung graft dysfunction and reduces the survival of these lung transplants [22].

For variable x28 (cold ischemia time = short) ischemia time had a number of harmful effects, especially for long duration times [23]. Most transplant centers try to limit cold

**Table 2. Expression of the probability equation associated to the logistic regression initial covariates and product units model.**

| Method | #param. | Best model |
|---|---|---|
| LRIPU | 48 | $\ln(p/(1-p)) = 2.50 - 4.83((x_1)^{1.46}(x_9)^{2.27}(x_{11})^{-1.76}(x_{14})^{3.67}(x_{15})^{0.37}(x_{23})^{6.03}(x_{24})^{3.15}(x_{26})^{0.06}(x_{29})^{3.96}(x_{33})^{1.45}))$ |
| | | $+0.30(x_1) - 2.16(x_2) + 0.16(x_3) - 0.44(x_4) + 0.64(x_5) + 0.25(x_6) - 0.17(x_7) + 0.55(x_8)$ |
| | | $-0.70(x_9) - 0.31(x_{10}) - 0.32(x_{11}) - 1.55(x_{12}) - 0.07(x_{13}) - 0.46(x_{14}) + 0.22(x_{15})$ |
| | | $+0.18(x_{16}) + 0.63(x_{17}) + 1.25(x_{18}) - 2.27(x_{19}) - 0.05(x_{20}) + 0.06(x_{21}) + 0.03(x_{22})$ |
| | | $+0.63(x_{23}) - 2.21(x_{24}) + 0.14(x_{25}) + 0.73(x_{26}) - 0.81(x_{27}) + 1.01(x_{28}) - 0.83(x_{29})$ |
| | | $-1.34(x_{30}) + 0.60(x_{31}) + 3.12(x_{32}) + 23.50(x_{33}) + 3.03(x_{34}) - 23.51(x_{35}) - 0.50(x_{36})$ |

LRIPU, logistic regression using initial covariates and product units

#param., number of parameters in the model; *p*: Probability of survival after six months. Variables are scaled in the [1, 2] range.

ischemia times to less than 8 hours to improve survival [22]. However, attempts are now being made to minimize the deleterious effects of cold ischemia by applying pulmonary 'ex vivo pulmonary perfusion' techniques, which have significantly improved in recent years. These ex vivo perfusion devices reduce ischemic injury by improving oxygenation, providing metabolic support to the organ, maintaining alveolar-capillary integrity, and minimizing the effect of edema. The advantages of these devices include the maintenance and improvement of the quality of the transplantation organs that have prolonged ischemia times. As the rate of transplanted lungs from donors with expanded criteria increases, the need for organs to be transplanted is alleviated, thus minimizing the incidence of early transplanted graft dysfunction [24].

Variables with negative coefficients negatively influenced the probability of graft survival after transplantation, such that the higher the absolute value of the coefficient, the lower the probability of survival. These variables were x2, x4, x7, x9, x10, x11, x12, x13, x14, x19, x20, x24, x27, x29, x30, x35, and x36, with x35 (fev1_pre) producing the greatest negative influence (with an absolute value of 23.51). The influence of this factor has mainly been observed in pulmonary allograft dysfunction where a decrease in the FEV1 value with respect to baseline values is related to dysfunction and primary graft rejection [19]. To a lesser degree, several studies have also related x19 (io2_donor, 2.27) to a lower probability of graft survival. Thus, in some isolated studies, a donor oxygenation index (OI2/FiO2) <300, has been related to graft failure [25–28], although most studies do not report an increased risk to the recipient [26, 28, 29].

In our work, the survival prognosis was poorer for recipients and older donors, and so x2 (age_rec, 2.16), the recipient age > 65 years was considered a relative contraindication for lung transplantation [30], although this age limit remains highly controversial [31, 32]. Older recipients generally had a shorter survival times than younger recipients, although in some studies these differences did not exist [31, 32]. Similarly, increased donor age (x12, age_donor, 1.55) worsened the prognosis, so that the risk of a lung transplant from an extremely elderly donor was uncertain and may lead to an increased risk [22]. As in our study, donor age> 55 years and prolonged ischemic times have also been associated with worse survival outcomes [33], and donor age> 50 years has been independently associated with decreased survival [34]. In cases of differences in donor and recipient lung size, and when Taylor reduction surgery was sometimes necessary, x30 (Taylor measure, 1.34), in accordance with other studies, this size discrepancy increased mortality in cases of single undersized lung transplantation [35].

The joint interaction between variables x1, x9, x11, x14, x15, x23, x24, x26, x29, and x33 was positively correlated with the probability of non-survival. Thus, the positive exponent variables

increased the product unit values and consequently, the probability of non-survival, with thex23 variable (type_tx = bilobular cadaver) showing the highest positive exponent of 6.03. The influence of the type of transplant on lung recipient survival is a highly controversial issue, which has been extensively studied. For example, a single lung transplant is associated with an increased risk of primary graft dysfunction and increased mortality [23], although other studies have not confirmed this association [36]. The probability of non-survival also increases when a cardiopulmonary bypass had been performed, x29 (bypass), with an exponent of 3.96, although this has been considered an independent factor of primary graft dysfunction and, therefore, increases the probability of death [22, 23, 36].

The cause of brain death, especially when cause by traumatic brain injury (x14, death_d = TBI) with an exponent of 3.67, also influenced the transplantation prognosis. The association between acute brain injury and subsequent pulmonary dysfunction is well recognized clinically. Patients suffering from massive irreversible brain injury, resulting in a diagnosis of brain stem death, also have a high incidence of associated pulmonary dysfunction [37]. However, in the broader analysis of TBI donors and the impact on lung transplant survival to date, no difference has been found over a 5-year period in lung transplant recipients from TBI donors versus non-TBI donors [38].

To a lesser degree, the clinical status of the patient prior to transplantation x9 (pre_tx = hospitalized, 2.27) also influenced the survival result; in line with this result, hospitalized and unstable patients requiring mechanical ventilation prior to transplantation had significantly worse survival rates [39]. In this study, female recipients also had a lower probability of survival, x1 (sex_rec, 1.46). Other studies have reported a similar association between female recipient sex and higher rates of primary graft dysfunction and increased mortality, albeit with significant heterogeneity in different patient populations [36]; however, still other studies have failed to identify this same recipient sex difference as independent risk factors for primary graft dysfunction or increased mortality [23]. For example, the FVC, x33 (fvc_pre, 1.45) was one of the factors that influenced survival both before and after lung transplantation [19].

However, donor sex, x11 (sex_donor) had an exponent of −1.76 which was negatively related to the product unit value, such that male donors had an increased probability of survival. Of note, the absolute value was quite low and so this influence may be overshadowed by interactions with the rest of the variables. The impact of donor sex presented contradictory results in different publications [40]. In a multicenter study, sex was not identified as an independent risk factor for increased primary lung dysfunction or increased in mortality [23]. The increased risk attributed to donor sex was probably related to the donor organ size being smaller than average size for the recipients' sex [22]. Male lungs are generally about 20% larger than female lungs [41], which may explain the probability differences we found in this work.

## Conclusions

Work and activities aimed at improving the problem of the shortage of organs for transplantation, should include measures to optimize donors suffering brain death, to promote live donation, evaluate expanded criteria for organ transplantation, and increase donation in asystole or with the used of ex vivo lung recovery machines. We believe that the model we obtained in this retrospective review of the donor-recipient lung allocation system may be a new way forward for optimizing the lung donation and transplantation process. This system should also consider additional variables of transplantation benefit, i.e., allocation considering survival and maximized use given the increased demand for organs which is encouraging the transplantation of organs by applying expanded criteria.

Artificial neural networks, based on data and variables resulting from the experience and results of multiple transplants, could play an important role in matching lung transplant donor-recipients, thus optimizing efficiency, effectiveness, and equity, and helping to avoid any subjectivity that may play a personal role in decision-making by professionals. The fusion of LR and neural networks can provide information on the importance of each of the non-linear relationships between variables that influence the evolution of lung transplantation. Finally, it would be interesting to validate these results, both internally and externally, and to carry out a prospective multicenter clinical trial validation to try to achieve greater robustness when applying the allocation model in lung the context of transplantation allocations.

## Supporting information

**S1 Data.**
(XLSX)

**S2 Data.**
(XLSX)

## Acknowledgments

We would like to thank all the dedicated staff at the Reina Sofia University Hospital, Córdoba, Spain.

## Author Contributions

**Conceptualization:** J. M. Dueñas-Jurado, P. A. Gutiérrez, A. Casado-Adam, F. Santos-Luna, A. Salvatierra-Velázquez, S. Cárcel, C. J. C. Robles-Arista, C. Hervás-Martínez.

**Data curation:** J. M. Dueñas-Jurado, P. A. Gutiérrez, A. Casado-Adam, F. Santos-Luna, A. Salvatierra-Velázquez, S. Cárcel, C. J. C. Robles-Arista, C. Hervás-Martínez.

**Formal analysis:** J. M. Dueñas-Jurado, P. A. Gutiérrez, A. Casado-Adam, F. Santos-Luna, A. Salvatierra-Velázquez, S. Cárcel, C. J. C. Robles-Arista, C. Hervás-Martínez.

**Funding acquisition:** J. M. Dueñas-Jurado, P. A. Gutiérrez, A. Casado-Adam, F. Santos-Luna, A. Salvatierra-Velázquez, S. Cárcel, C. J. C. Robles-Arista, C. Hervás-Martínez.

**Investigation:** J. M. Dueñas-Jurado, P. A. Gutiérrez, A. Casado-Adam, F. Santos-Luna, A. Salvatierra-Velázquez, S. Cárcel, C. J. C. Robles-Arista, C. Hervás-Martínez.

**Methodology:** J. M. Dueñas-Jurado, P. A. Gutiérrez, A. Casado-Adam, F. Santos-Luna, A. Salvatierra-Velázquez, S. Cárcel, C. J. C. Robles-Arista, C. Hervás-Martínez.

**Project administration:** J. M. Dueñas-Jurado, P. A. Gutiérrez, A. Casado-Adam, F. Santos-Luna, A. Salvatierra-Velázquez, S. Cárcel, C. J. C. Robles-Arista, C. Hervás-Martínez.

**Resources:** J. M. Dueñas-Jurado, P. A. Gutiérrez, A. Casado-Adam, F. Santos-Luna, A. Salvatierra-Velázquez, S. Cárcel, C. J. C. Robles-Arista, C. Hervás-Martínez.

**Software:** J. M. Dueñas-Jurado, P. A. Gutiérrez, A. Casado-Adam, F. Santos-Luna, A. Salvatierra-Velázquez, S. Cárcel, C. J. C. Robles-Arista, C. Hervás-Martínez.

**Supervision:** J. M. Dueñas-Jurado, P. A. Gutiérrez, A. Casado-Adam, F. Santos-Luna, A. Salvatierra-Velázquez, S. Cárcel, C. J. C. Robles-Arista, C. Hervás-Martínez.

**Validation:** J. M. Dueñas-Jurado, P. A. Gutiérrez, A. Casado-Adam, F. Santos-Luna, A. Salvatierra-Velázquez, S. Cárcel, C. J. C. Robles-Arista, C. Hervás-Martínez.

**Visualization:** J. M. Dueñas-Jurado, P. A. Gutiérrez, A. Casado-Adam, F. Santos-Luna, A. Salvatierra-Velázquez, S. Cárcel, C. J. C. Robles-Arista, C. Hervás-Martínez.

**Writing – original draft:** J. M. Dueñas-Jurado, P. A. Gutiérrez, A. Casado-Adam, F. Santos-Luna, A. Salvatierra-Velázquez, S. Cárcel, C. J. C. Robles-Arista, C. Hervás-Martínez.

**Writing – review & editing:** J. M. Dueñas-Jurado, P. A. Gutiérrez, A. Casado-Adam, F. Santos-Luna, A. Salvatierra-Velázquez, S. Cárcel, C. J. C. Robles-Arista, C. Hervás-Martínez.

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
