## [Decision Letter · Decision Letter 0]

21 Oct 2020

PONE-D-20-28350

New models for matching donor-recipient in lung transplantations in the current COVID-19 pandemic

PLOS ONE

Dear Dr. Casado-Adam,

Thank you for submitting your manuscript to PLOS ONE. After careful consideration, we feel that it has merit but does not fully meet PLOS ONE’s publication criteria as it currently stands. Therefore, we invite you to submit a revised version of the manuscript that addresses the points raised during the review process.

Please revise accordingly.

We look forward to receiving your revised manuscript.

Kind regards,

Academic Editor

PLOS ONE

Journal Requirements:

2) We note that you have indicated that data from this study are available upon request. PLOS only allows data to be available upon request if there are legal or ethical restrictions on sharing data publicly. For more information on unacceptable data access restrictions, please see http://journals.plos.org/plosone/s/data-availability#loc-unacceptable-data-access-restrictions.

3) PLOS requires an ORCID iD for the corresponding author in Editorial Manager on papers submitted after December 6th, 2016. Please ensure that you have an ORCID iD and that it is validated in Editorial Manager. To do this, go to ‘Update my Information’ (in the upper left-hand corner of the main menu), and click on the Fetch/Validate link next to the ORCID field. This will take you to the ORCID site and allow you to create a new iD or authenticate a pre-existing iD in Editorial Manager. Please see the following video for instructions on linking an ORCID iD to your Editorial Manager account: https://www.youtube.com/watch?v=_xcclfuvtxQ

Reviewers' comments:

Reviewer's Responses to Questions

**Comments to the Author**

1. Is the manuscript technically sound, and do the data support the conclusions?

Reviewer #1: Yes

Reviewer #2: Yes

2. Has the statistical analysis been performed appropriately and rigorously? 

Reviewer #1: I Don't Know

Reviewer #2: I Don't Know

3. Have the authors made all data underlying the findings in their manuscript fully available?

Reviewer #1: Yes

Reviewer #2: Yes

4. Is the manuscript presented in an intelligible fashion and written in standard English?

Reviewer #1: Yes

Reviewer #2: No

5. Review Comments to the Author

Reviewer #1: I commend the authors on their fine work. With COVID, the allocation of lungs is even more difficult.

Please review for grammar and english syntax. There are multiple grammatically incorrect sentences in the introduction. It seems very disjointed and I cannot go over the whole thing in this review. Also I think the authors meant for incomplete sentences to be opening sentences of paragraphs. but they just appear as incomplete sentences.

From reading the introduction, it seems like the authors are using COVID as a buzz word. Besides the fact that COVID decreased transplants, how else is it affecting lung transplantation. How have institutions dealt with this? There should be more regarding COVID and transplantation if the authors want to bring up COVID.

As for the methods section, I have asked the editors to have a statistician review the statistics as I am not well versed in neural netorks.

With the results of the study, please draw some conclusions. The conclusion section is basically a restatement of all the results, which is inappropriate for a conclusion. Also, if the authors would like to discuss COVID and transplant, please review this in the conclusion. No where in the conclustions is there anything about COVID.

Reviewer #2: This isnq retrospective review on f a Spanish lung transplant registry data which is evaluated for new paradigms

First and foremost you often state that this is a problem relative to COVID which it is not. The data are from before Jan 2016 well before any COVID infection and the data is never applied t patients undergoing transplant during the COVID era. Please remove all references to COVID in the title and most other places

What about perfusion circuits and their effect on the variable of cold ischemia?

You imply that this will help with organ shortage? How would that happen!?l

Did yu plan a multicenter preferably international validation study?

English good ammar needs upgrading

6. PLOS authors have the option to publish the peer review history of their article (what does this mean?). If published, this will include your full peer review and any attached files.

Reviewer #1: No

Reviewer #2: No

---

## [Author Response · Author response to Decision Letter 0]

8 Feb 2021

Thank you Reviewers for the comments. We hope the corrections are to your liking.

---

## [Decision Letter · Decision Letter 1]

3 Mar 2021

PONE-D-20-28350R1

New models for donor-recipient matching in lung transplantations

PLOS ONE

Dear Dr. Casado-Adam,

Thank you for submitting your manuscript to PLOS ONE. After careful consideration, we feel that it has merit but does not fully meet PLOS ONE’s publication criteria as it currently stands. Therefore, we invite you to submit a revised version of the manuscript that addresses the points raised during the review process.

Please revise accordingly.

We look forward to receiving your revised manuscript.

Kind regards,

Academic Editor

PLOS ONE

Reviewers' comments:

Reviewer's Responses to Questions

**Comments to the Author**

1. If the authors have adequately addressed your comments raised in a previous round of review and you feel that this manuscript is now acceptable for publication, you may indicate that here to bypass the “Comments to the Author” section, enter your conflict of interest statement in the “Confidential to Editor” section, and submit your "Accept" recommendation.

Reviewer #1: (No Response)

Reviewer #3: All comments have been addressed

2. Is the manuscript technically sound, and do the data support the conclusions?

Reviewer #1: Yes

Reviewer #3: Yes

3. Has the statistical analysis been performed appropriately and rigorously? 

Reviewer #1: Yes

Reviewer #3: Yes

4. Have the authors made all data underlying the findings in their manuscript fully available?

Reviewer #1: Yes

Reviewer #3: Yes

5. Is the manuscript presented in an intelligible fashion and written in standard English?

Reviewer #1: (No Response)

Reviewer #3: Yes

6. Review Comments to the Author

Reviewer #1: This revision clearly has changes in it but I cannot tell without a copy of track changes. Also, in your responses, there is no point by point response, just a thank you for the comments. This is very sparse. It used to be a COVID lung transpalnt paper, and now its just an analysis of the Spanish registry. Please submit a trach changes so I can tell what is new. If not, then since this has been changed from a "COVID" paper to this, please submit a new submission.

Reviewer #3: Dear The Authors

Thank you for writing this manuscript

I have couple of points to mention here:

1. Please run a second round of English language spell check for the whole manuscript

2. Please remove the part pertaining to COVID-19 since your study period is covering before the pandemic

Thank you

7. PLOS authors have the option to publish the peer review history of their article (what does this mean?). If published, this will include your full peer review and any attached files.

Reviewer #1: No

Reviewer #3: **Yes: **salah eldien Altarabsheh

---

## [Author Response · Author response to Decision Letter 1]

20 Apr 2021

Initially, the work was designed prior to the appearance of COVID, although we think it could be interesting to include it to give originality to our work, we recognize its inclusion may be somewhat artificial at this time.

In accordance with your comments on the appearance of COVID terms, and also following the recommendations of our other reviewer, we have removed everything related to COVID from our text. We have tried following your directions to seek general benefits in the lung donation and transplantation program based on the ability of the machines to learn independently and make accurate predictions. We not only treat a Spanish registry analysis, but rather we try to apply artificial intelligence combined with the “big data” of electronic medical records to achieve the highest precision when matching donor-recipient organs and optimize our results.

---

## [Decision Letter · Decision Letter 2]

11 May 2021

New models for donor-recipient matching in lung transplantations

PONE-D-20-28350R2

Dear Dr. Casado-Adam,

We’re pleased to inform you that your manuscript has been judged scientifically suitable for publication and will be formally accepted for publication once it meets all outstanding technical requirements.

Kind regards,

Academic Editor

PLOS ONE

Additional Editor Comments (optional):

Reviewers' comments:

Reviewer's Responses to Questions

**Comments to the Author**

1. If the authors have adequately addressed your comments raised in a previous round of review and you feel that this manuscript is now acceptable for publication, you may indicate that here to bypass the “Comments to the Author” section, enter your conflict of interest statement in the “Confidential to Editor” section, and submit your "Accept" recommendation.

Reviewer #1: All comments have been addressed

Reviewer #3: All comments have been addressed

2. Is the manuscript technically sound, and do the data support the conclusions?

Reviewer #1: Yes

Reviewer #3: Yes

3. Has the statistical analysis been performed appropriately and rigorously? 

Reviewer #1: I Don't Know

Reviewer #3: Yes

4. Have the authors made all data underlying the findings in their manuscript fully available?

Reviewer #1: Yes

Reviewer #3: Yes

5. Is the manuscript presented in an intelligible fashion and written in standard English?

Reviewer #1: (No Response)

Reviewer #3: Yes

6. Review Comments to the Author

Reviewer #1: Thank you for addressing my comments. The COVID aspect of the prior paper definitely made it more distracting and I couldn't understand it as well. It is better understood now.

Reviewer #3: Thank you for taking my comments into consideration

I have no concerns about this manuscript in its current form

7. PLOS authors have the option to publish the peer review history of their article (what does this mean?). If published, this will include your full peer review and any attached files.

Reviewer #1: No

Reviewer #3: **Yes: **Salah Altarabsheh

---

## [Editor Report · Acceptance letter]

26 May 2021

PONE-D-20-28350R2 

New models for donor-recipient matching in lung transplantations 

Dear Dr. Casado-Adam:

I'm pleased to inform you that your manuscript has been deemed suitable for publication in PLOS ONE. Congratulations! Your manuscript is now with our production department. 

Kind regards, 

on behalf of

Dr. Robert Jeenchen Chen 

Academic Editor

PLOS ONE